# SBDS^R126T rescues survival of *sbds*^−/− zebrafish in a dose-dependent manner independently of Tp53

Usua Oyarbide[1] , Arish N Shah[2] , Morgan Staton[1] , Matthew Snyderman[1], Adya Sapra[1], Eliezer Calo[2] , Seth J Corey[1]

**Defects in ribosomal biogenesis profoundly affect organismal development and cellular function, and these ribosomopathies produce a variety of phenotypes. One ribosomopathy, Shwachman–Diamond syndrome (SDS) is characterized by neutropenia, pancreatic exocrine insufficiency, and skeletal anomalies. SDS results from biallelic mutations in *SBDS*, which encodes a ribosome assembly factor. Some individuals express a missense mutation, *SBDS*^R126T, along with the common K62X mutation. We reported that the *sbds*-null zebrafish phenocopies much of SDS. We further showed activation of Tp53-dependent pathways before the fish died during the larval stage. Here, we expressed *SBDS*^R126T as a transgene in the *sbds*^−/− background. We showed that one copy of the *SBDS*^R126T transgene permitted the establishment of maternal zygotic *sbds*-null fish which produced defective embryos with *cdkn1a* up-regulation, a Tp53 target involved in cell cycle arrest. None survived beyond 3 dpf. However, two copies of the transgene resulted in normal development and lifespan. Surprisingly, neutropenia persisted. The surviving fish displayed suppression of female sex differentiation, a stress response in zebrafish. To evaluate the role of Tp53 in the pathogenesis of *sbds*^−/− fish phenotype, we bred the fish with a DNA binding deficient allele, *tp53*^M214K. Expression of the loss-of-function *tp53*^M214K did not rescue neutropenia or survival in *sbds*-null zebrafish. Increased expression of *cdkn1a* was abrogated in the *tp53*^M214K/M214K;*sbds*^−/− fish. We conclude that the amount of SBDS^R126T protein is important for development, inactivation of Tp53 fails to rescue neutropenia or survival in the *sbds*-null background, and *cdkn1a* up-regulation was dependent on WT *tp53*. We hypothesize that additional pathways are involved in the pathophysiology of SDS.**

# Introduction

Shwachman–Diamond syndrome (SDS) is an autosomal recessive disorder characterized by neutropenia, pancreatic insufficiency, and skeletal defects. It confers an increased risk of transforming to a myeloid neoplasm, either myelodysplastic syndrome or acute myeloid leukemia in 15–25% of affected individuals (Donadieu et al, 2005; Dror, 2005). Almost all SDS cases are because of mutation in the Shwachman–Bodian–Diamond syndrome (*SBDS*) gene (Boocock et al, 2003). SBDS physically interacts with the GTPase elongation factor-like 1 (EFL1) to release the eukaryotic-initiating factor 6 (EIF6) from the cytoplasmic pre-60S ribosomal subunit. This release facilitates the assembly of the mature 80S ribosome (Finch et al, 2011).

Biallelic mutations in *SBDS* account for ~90% of patients (Boocock et al, 2003). Most commonly, *SBDS* mutations are located in exon 2 and lead to disruption of a donor splice and a frameshift mutation (C83fs) or protein truncation after introduction of a stop codon (K62X). C83fs mutation produces reduced expression of full-length protein (Austin et al, 2005). Whereas a few patients are homozygous for the splice donor mutation, homozygous mutants for K62X have not been identified, suggesting that complete loss of SBDS is lethal (Boocock et al, 2003; Austin et al, 2005; Shammas et al, 2005). Moreover, gene ablation of *Sbds* in mice results in early embryonic lethality (E 6.5) (Zhang et al, 2006a). The SDS-associated missense mutation, R126T, is hypomorphic, unable to activate the GTPase activity of the EFL1, and inhibits the release of EIF6 from the 60S particle (Finch et al, 2011; Weis et al, 2015).

The small number of patients, their phenotypic diversity, and a long latency period to disease complications makes SDS difficult to study. Animal models thus play an essential role in identifying its pathogenesis and development of new treatments. The Rommens' group created a mouse model with a point mutation *c.377G>C* (p.R126T) in the *Sbds* allele. The *Sbds*^R126T/R126T and *Sbds*^R126T/− mice did not survive up to birth (Zhang et al, 2006a, 2006b; Tourlakis et al, 2012, 2015). Interestingly, *Sbds*^R126T/− mice were smaller and had decreased bone marrow cellularity, which was more severe than in their *Sbds*^R126T/R126T littermates. Pancreas-specific *Sbds* KO mice presented severe atrophy of the acinar component of the adult pancreas that was Tp53-dependent. This pancreatic atrophy was alleviated through total ablation of Tp53 (i.e., Trp53^−/−mouse) (Tourlakis et al, 2015).

We created zebrafish *sbds* KO strains that phenocopy the human syndrome with neutropenia, pancreatic atrophy, and small size. Polysome analysis showed decreased 80S ribosomes and

---

[1]Departments of Cancer Biology and Pediatrics, Cleveland Clinic, Cleveland, OH, USA [2]Department of Biology and David H. Koch Institute for Integrative Cancer Research, Massachusetts Institute of Technology, Cambridge, MA, USA

Correspondence: oyarbiu@ccf.org

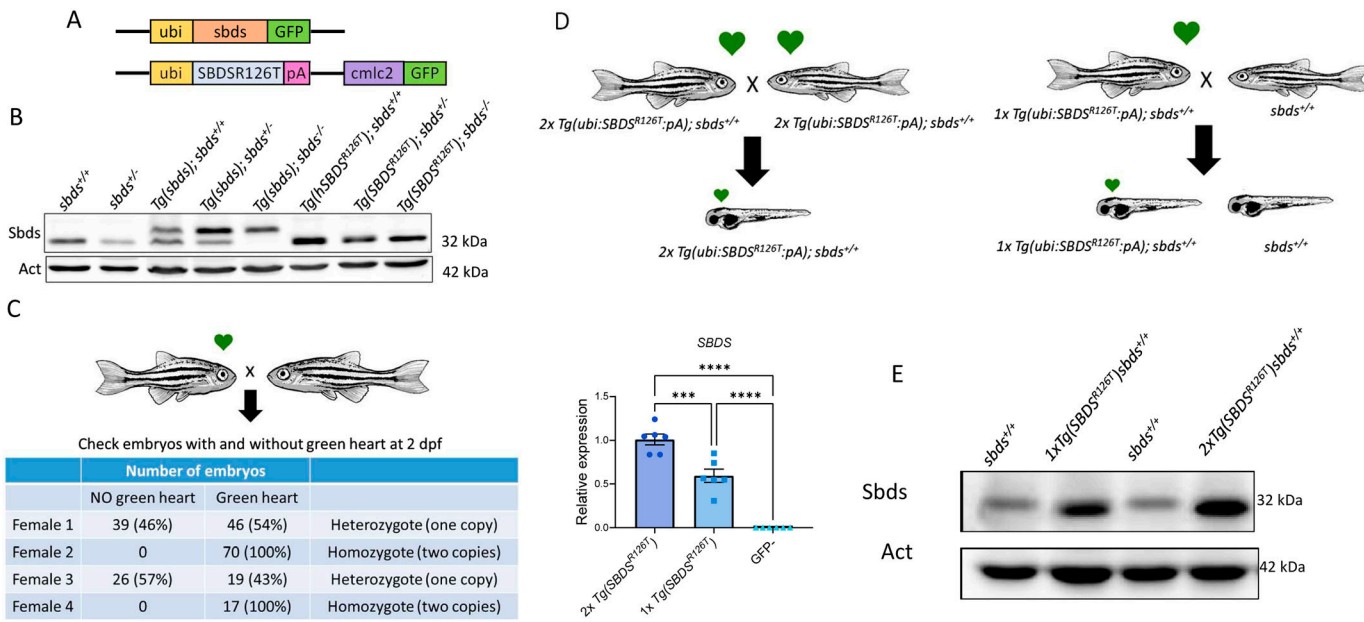

**Figure 1. Generation and characterization of the zebrafish transgenic strain expressing *SBDS^R126T*.**
**(A)** Constructs to create the transgenic line expressing zebrafish sbds and human SBDS^R126T were injected into fertilized eggs and, subsequently, strains were inbred. **(B)** Western blotting showed Sbds/SBDS^R126T expression in zebrafish fins on 1-yr-old fish. **(C)** The number of transgene copies was determined in the zebrafish SBDS^R126T line. **(D)** mRNA levels of human *SBDS* in 2 dpf larvae with one and two copies of the transgene *Tg(SBDS^R126T)*. **(E)** Immunoblot showing protein levels in fish with one and two copies of the transgene comparing with WT *sbds*. ubi-ubiquitin promoter, *cmlc2* green heart marker to aid screening of transgenesis.

accumulation of pre-60S ribosomal large subunit. RNA-seq with qRT-PCR validation revealed activation of tp53-associated pathways at 10 days post fertilization (dpf). The *sbds*-null fish died between 15–21 dpf (Oyarbide et al, 2020). These mutants represent an opportunity to explore the phenotypic consequences of the human disease-associated *SBDS^R126T* mutation. Here, we report a novel zebrafish transgenic line expressing the *SBDS^R126T* mutation under the *ubiquitin* (*ubi*) promoter that in high doses (2x copies of the transgene) rescued the *sbds* KO early mortality and neutropenia but did not abrogate activation of the tp53/cdkn1a pathway. We also explored the effect of *tp53^M214K* mutation, which is a loss-of-function in the DNA-binding domain (Berghmans et al, 2005), in the *sbds* mutant backgrounds, and determined that Tp53 inactivation is not sufficient for their survival or neutropenia.

## Results

### SBDS^R126T rescued the sbds^−/− zebrafish

We recently reported that *sbds*-null fish died between 15–21 dpf. We generated a zebrafish transgenic line expressing the zebrafish WT *sbds* under the zebrafish ubiquitin (*ubi*) promoter: *Tg(ubi:sbds:pA)*, which drives constitutive transgene expression during all developmental stages and adult organs (Mosimann et al, 2011). This transgenic line in the *sbds* KO background was able to rescue their phenotype and was viable (Oyarbide et al, 2020). To determine if expression of SBDS^R126T could prolong survival and modify aberrant development, we created a transgenic strain *Tg(ubi:SBDS^R126T:pA)* and bred that against the null background (also denoted

*Tg(SBDS^R126T)* here). We detected full-length SBDS protein (Fig 1A and B). Unexpectedly, the transgenic line *Tg(ubi:SBDS^R126T:pA)* in the background of the *sbds* KO can live for >18 mo (adulthood). We used *cmlc2*:EGFP as a transgenesis marker in the vector backbone, which drives cytoplasmic EGFP specifically in the heart and facilitates the screening for the presence of the transgene (Kwan et al, 2007).

To determine the number of copies of the transgene inserted in the genome, we crossed two siblings with green hearts and selected four females with green hearts from the descendants. We then outcrossed them with non-green heart fish and counted the number of green heart fish versus non-green heart fish. Two females showed a 100% of the descendants with green heart, whereas the other two had ~50% with green heart and 50% with non-green heart. We concluded that there was only one insertion of the transgene in the new transgenic line created (Fig 1C).

Next, we determined the human *SBDS^R126T* mRNA levels in 2 dpf larvae with 2x and 1x copies of the transgene. We incrossed 2x *Tg(ubi:SBDS^R126T:pA);sbds^+/+* and collect larvae at 2 dpf. In parallel, we outcrossed 1x *Tg(ubi:SBDS^R126T:pA);sbds^+/+* with a *sbds^+/+* and screened fish for green heart and non-green heart at 2 dpf. Next, we calculated the expression of *SBDS* in these fish and we observed a significant decrease of approximately half of the levels in the 1x comparing with 2x copies. As expected, those without a green heart showed no expression of human *SBDS* (Fig 1D). We also determined the protein levels in adult fish fins and observed an increase in protein levels in the 2x comparing with the 1x copy of the transgene (Fig 1E).

We then studied the phenotype of the adult fish at 1 yr. We crossed a *sbds^+/−* with a *Tg(ubi:SBDS^R126T:pA);sbds^+/−* (Fig 2A). This cross generated siblings from the same clutch without green heart

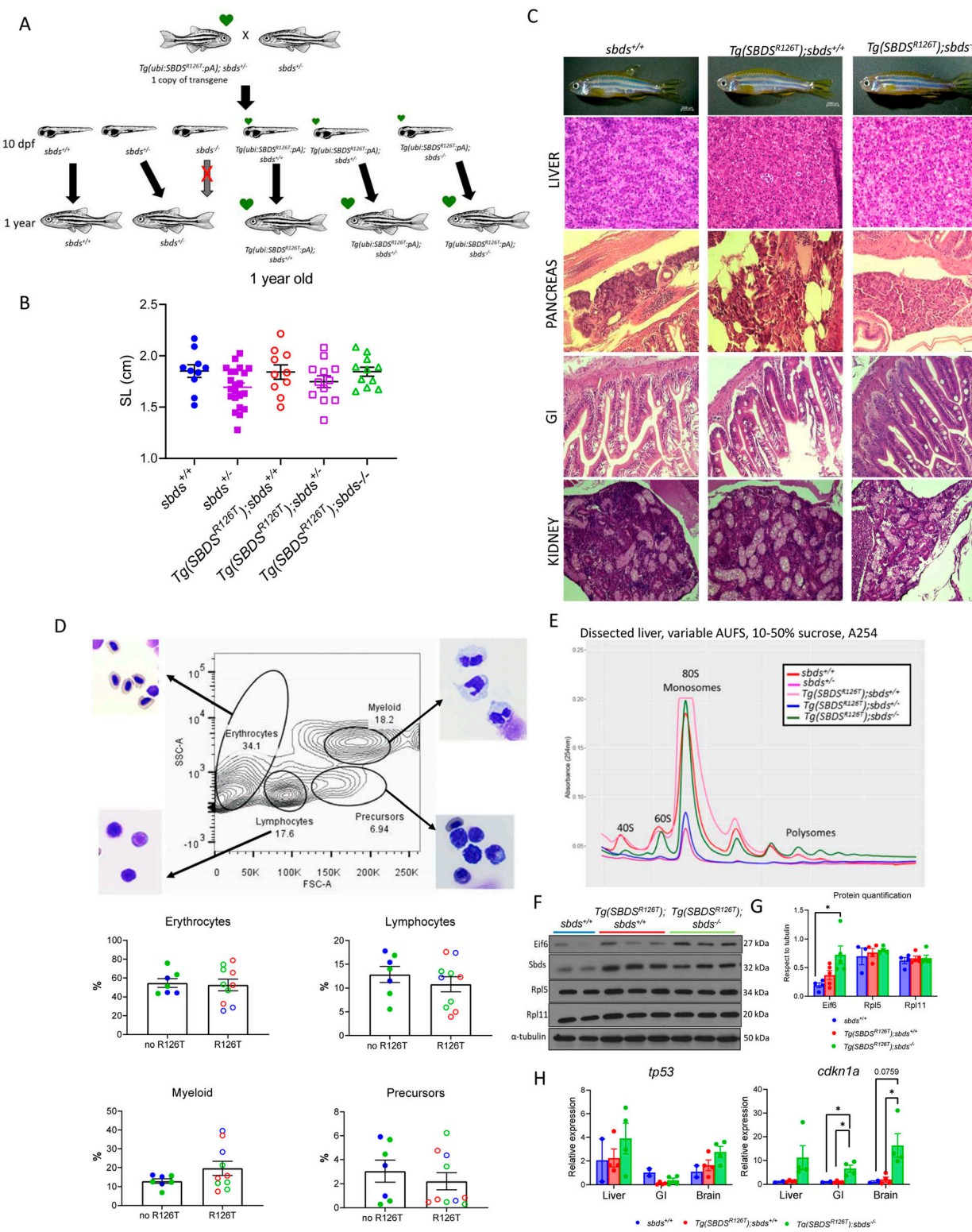

**Figure 2.   Phenotype and biochemical analysis of adult zebrafish lines expressing the human SBDS[R126T].**
**(A)** Zebrafish cross between *sbds* heterozygous and transgenic line. **(B)** Standard length in 1-yr-old fish. **(C)** H&E staining of internal organs of 1-yr-old male fish. **(D)** Flow cytometric analysis of whole kidney marrows showing the different blood cell types in the presence or absence of the transgene *SBDS[R126T]*, in the different zebrafish *sbds* backgrounds. **(E)** Polysome profile of the liver. **(F)** Western blotting demonstrated Eif6 accumulation in zebrafish *sbds* KO with the transgene. **(G, H)** Protein quantification (H) qRT-PCR analysis of liver, brain, and GI showing up-regulation of *cdkn1a* in all tissues. AUFS, absorbance units full scale.

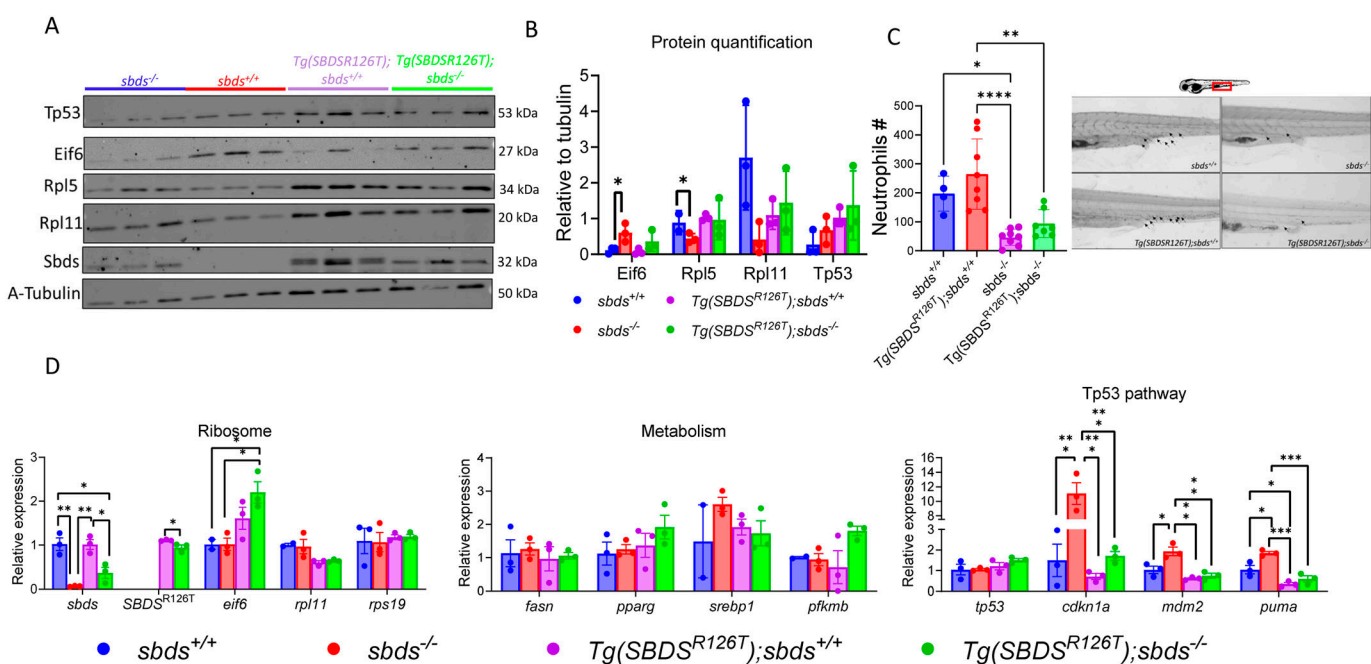

**Figure 3. Phenotype of 10 dpf larvae expressing the human SBDS^R126T.**
**(A)** Western blot showed Eif6 without RPL5, and Rpl11 accumulation in the sbds KO fish, but there was no Eif6 accumulation in the SBDS^R126T strain. **(B)** Protein quantification of Western blots as performed with NIH ImageJ. **(C)** After Sudan black staining, neutrophils were counted and found to be significant lower in the sbds KO background with or without the transgene. **(D)** qRT-PCR analysis of critical genes involved in ribosomal, metabolic, and tp53-associated pathways.

and with one copy of the transgene. Because the $sbds^{-/-}$ died between 10–21 dpf, we analyzed the other five phenotypes produced from this cross at 1 yr. We did not see differences in standard length (SL) or in the histologic sections including liver, pancreas, digestive tract or kidney (Fig 2B and C). Surprisingly, we observed a 7:1 male: female ratio in the $Tg(ubi:SBDS^{R126T}:pA)$ with the sbds-null background, and reduced fecundity. Flow cytometric profile of hematopoietic cells prepared from 1-yr-old kidney marrow of male siblings showed no difference in any of the blood cell populations (erythrocytes, lymphocytes, myeloid, and precursors) of the five different genotypes analyzed (Fig 2D).

Because 80S ribosome formation is affected in SDS patients and animal models including zebrafish (Finch et al, 2011; Tourlakis et al, 2012; Oyarbide et al, 2020), we performed polysome profiling on the livers of 1-yr-old male fish. Compared with what we had observed in the sbds-null fish, the expression of one transgenic copy of the $SBDS^{R126T}$ allele rescued the polysome profile to mostly normal (Fig 2E).

We previously reported that $sbds^{-/-}$ had a decrease in Rpl5 and Rpl11 protein levels and an accumulation of Eif6 protein levels (Oyarbide et al, 2020). We evaluated the levels of these proteins by Western blotting of fin lysates from the transgenic adult fish (1 yr-old). The ribosomal protein levels were similar to those found in WT siblings. However, Eif6 protein levels were significantly increased in the $Tg(ubi:SBDS^{R126T}:pA)$ $sbds^{-/-}$ (Fig 2F and G), as we previously reported in sbds-null larvae at 10 dpf (Oyarbide et al, 2020). We also analyzed the mRNA levels of tp53 and cdkn1a in liver, intestine, and brain; cdkn1a was markedly up-regulated in all three organs (Fig 2H).

Because we observed accumulation of Eif6 and activation of Tp53-pathway in the 1-yr-old transgenic fish, we evaluated these

pathways at earlier stages of development. We crossed a $sbds^{+/-}$ with $Tg(ubi:SBDS^{R126T}:pA)$ $sbds^{+/-}$ (Fig 2A) and analyzed the larvae at 10 dpf. At this stage, the $sbds^{-/-}$ were alive and were included in the analysis. Western blotting showed a statistically significant increase in Eif6 protein in the sbds-null fish as previously observed and a nonsignificant increase in the transgenic line with the sbds-null background. Rpl5 and Rpl11 decreased only in $sbds^{-/-}$ in comparison with the WT siblings (Fig 3A and B). Sudan black staining showed a decrease in the number of neutrophils in sbds KO background fish, independently of the presence of SBDS^R126T (Fig 3C).

To understand how the $SBDS^{R126T}$ allele affected the zebrafish larval development, we performed qRT-PCR to determine changes in gene expression in ribosomes (rpl11 and rps19), metabolism (fasn, pparg, srebp1, and pfkmb), and the tp53 pathway (tp53, cdkn1a, mdm2, and puma) (Fig 3D). As expected, sbds was down-regulated in $sbds^{-/-}$ and $Tg(ubi:SBDS^{R126T}:pA)$ $sbds^{-/-}$. Surprisingly, eif6 mRNA was up-regulated only in the $Tg(ubi:SBDS^{R126T}:pA)$ $sbds^{-/-}$, whereas rpl11 and rpl5 transcripts were not changed. We did not observe changes in the metabolism markers tested. We did find Tp53 pathway activation in the sbds KO through the up-regulation of cdkn1a, mdm2, and puma. These markers were not affected in the transgenic line of neither $sbds^{+/+}$ nor $sbds^{-/-}$ backgrounds (Fig 3D).

## Levels of $SBDS^{R126T}$ affected embryonic development in the $sbds^{-/-}$ fish

Hypothesizing that levels of SBDS^R126T might modify the embryonic development and severity of SDS phenotype, we incrossed

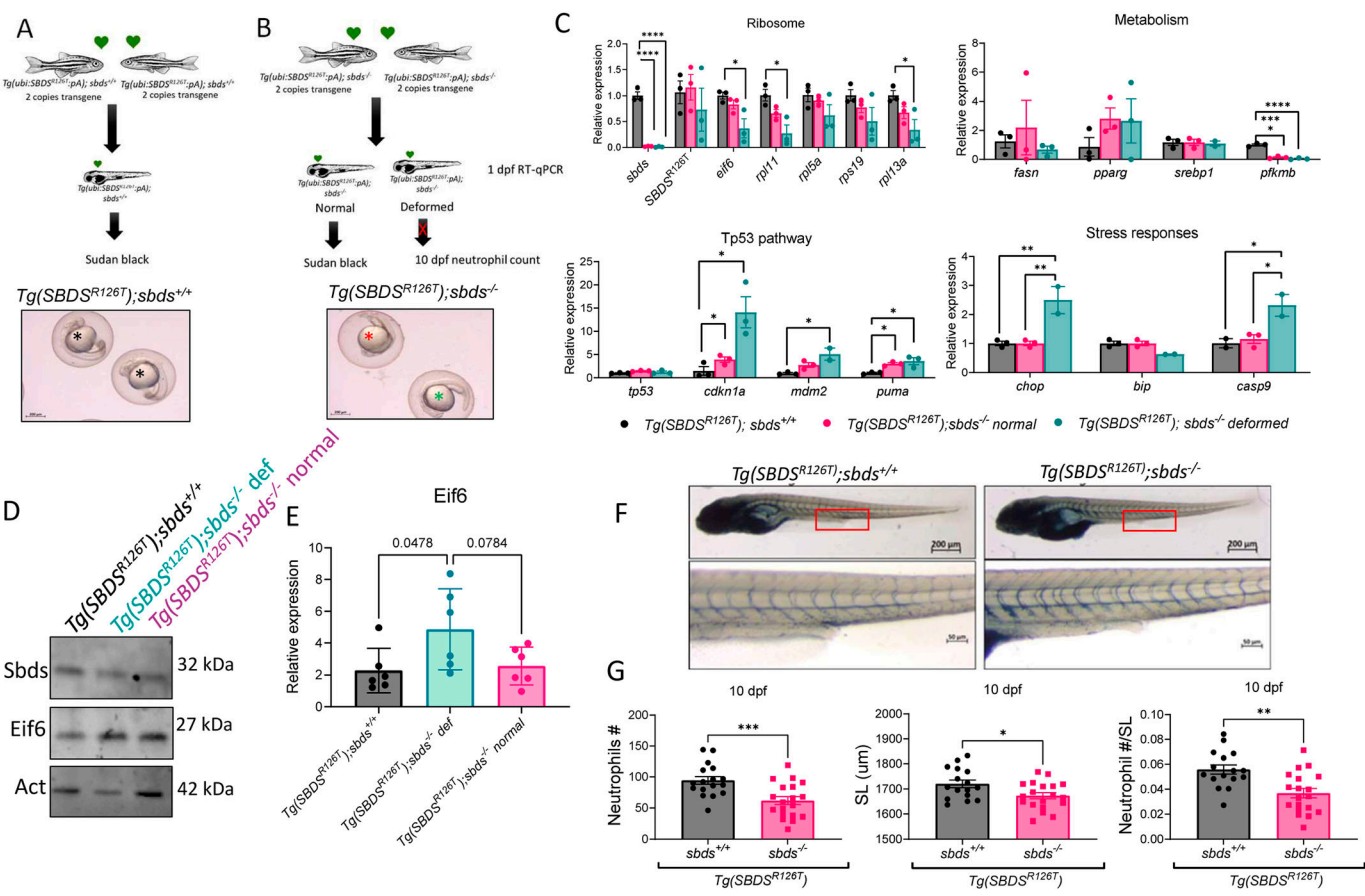

**Figure 4. Level of SBDS^R126T protein expression affected embryonic development.**
**(A)** Incross between WT sbds fish with two copies of the transgenic line produced healthy embryos. **(B)** Incross between sbds KO fish with two copies of the transgenic line produced healthy embryos and deformed embryos. Black star shows healthy embryo, green star shows developmentally delayed, and red star shows deformed embryo. **(C)** qRT-PCR analysis of critical genes involved in ribosomal, metabolic, and tp53-associated pathways. **(D)** Western blot showing Eif6 accumulation in zebrafish-deformed embryos. **(E)** Protein quantification of Western blots as performed with NIH ImageJ. **(F)** Sudan black staining of 10 dpf larvae to count neutrophils. **(G)** Standard length and neutrophil number were reduced in sbds KO larvae expressing SBDS^R126T.

*Tg(ubi:SBDS^R126T:pA);sbds^+/+* with two copies of the transgene so that all the descendants were *Tg(ubi:SBDS^R126T:pA);sbds^+/+* (Fig 4A). In parallel, we incrossed *Tg(ubi:SBDS^R126T:pA);sbds^−/−*, to have all the descendants with the same genotype *Tg(ubi:SBDS^R126T:pA);sbds^−/−* (maternal zygotic mutants [MZ]) (Fig 4B). Surprisingly, after 1 dpf we observed developmental delay in some embryos and deformed embryos in the *sbds^−/−* background (Fig 4A and B). To characterize the genetic and phenotypic differences, we collected embryos at either 1 dpf for gene expression analysis or embryos that developed normal at 10 dpf to measure neutrophil numbers. As expected, we found a significant down-regulation of *sbds* in the *sbds^−/−* (Fig 4C). However, *eif6* and all ribosomal proteins tested (*rpl11*, *rpl5a*, *rps19*, and *rpl13a*) were significantly down-regulated only in the deformed embryos. We previously showed that mRNA levels of one important enzyme in the pathway of glycolysis, phosphofructokinase (*pfkmb*), was down-regulated in the *sbds^−/−* fish (Oyarbide et al, 2020). We evaluated the *pfkmb* mRNA levels in the *Tg(SBDS^R126T)sbds^−/−*, which were decreased in the normal and deformed larvae comparing with *Tg(SBDS^R126T)sbds^+/+*. We did not see any change in the lipid metabolism markers. Next, we

analyzed Tp53 pathway: *cdkn1a* and *puma* were up-regulated in all the *sbds*-null backgrounds and *mdm2* in the deformed ones (Fig 4C). We also checked markers for stress response, where we found a significant up-regulation of *chop* and *casp9* in the sbds KO background (Fig 4C). We then checked Eif6 protein levels at 2 dpf in normal and deformed fish with the *sbds*-null background and compared the levels with the WT *sbds* background. Eif6 levels were increased only in the deformed embryos (Fig 4D and E).

Almost all of the normal-appearing *Tg(ubi:SBDS^R126T:pA);sbds^−/−* and the WT fish survived until 10 dpf, (respectively, 80% and 76%). However, *Tg(ubi:SBDS^R126T:pA);sbds^−/−* were significantly smaller and had significantly lower number of neutrophils comparing with the WT group (Fig 4F and G). All developed into males.

Next, we determined the effects of one copy of the transgene *SBDS^R126T* (Fig 5A). Interestingly, embryonic development was defective in 25% of the MZ *sbds^−/−* embryos after 1 dpf, the mortality was 100% after 3 dpf (Fig 5B and C), and presence of transgene did not ameliorate the defects (Fig 5B). We collected samples at 2 dpf (i.e., at the appearance of the green heart), and determined changes in gene expression. Surprisingly, *cdkn1a* was only up-regulated in

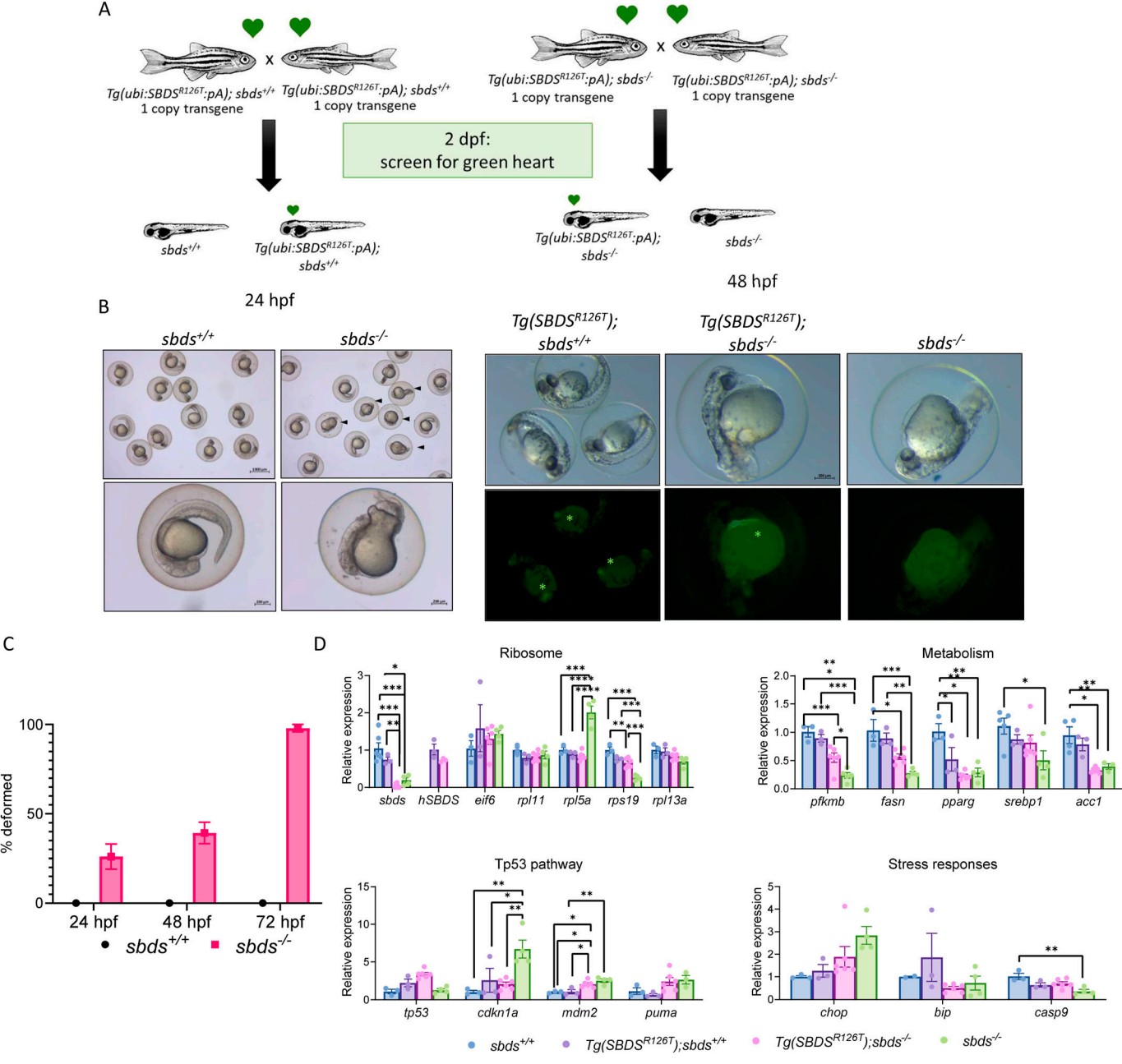

**Figure 5. Incrosses of *sbds* KO with one copy of the transgene have severe malformations in early stages of development.**
**(A)** Crosses between WT and sbds KO fish. **(B)** Embryos obtained from crosses at 24 and 48 hpf. **(C)** Percentage of deformed fish in the first 72 hpf. **(D)** qRT-PCR analysis of critical genes involved in ribosomes, metabolism, and tp53 pathways.

*sbds*-null embryos without the transgene. However, *pfkmb*, *fasn*, and *pparg* were down-regulated in *sbds*-null embryos with and without *SBDS^R126T^* (Fig 5D).

Previously, we reported that *sbds* null fish had fewer neutrophils than WT siblings at 5 dpf (Oyarbide et al, 2020). To determine whether the SBDS^R126T^ rescued the neutropenia in the *sbds*-null background, we incrossed *sbds* heterozygotes with two copies of the transgene and determined the number of neutrophils at 5 dpf (Fig 6A). As expected, we did not detect differences in neutrophil counts between any of the genotypes (*sbds^+/+^*, *sbds^+/-^*, and *sbds^-/-^*)

in the context of the transgenic *SBDS R126T* (Fig 6B). With these results, we can conclude that the SBDS^R126T^ dose is important in the neutrophil number in our zebrafish SDS models.

## Tp53-loss does not rescue neutropenia or survival of *sbds* mutants

The p53 tumor suppressor pathway is activated by impairment of ribosome biogenesis and aberrant protein translation (Narla &

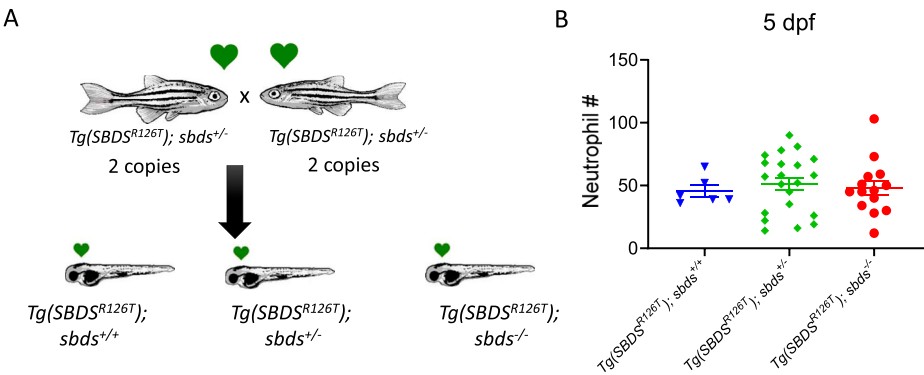

**Figure 6.   Two copies of the *SBDS^{R126T}* transgene rescues neutropenia at 5 dpf.**
**(A)** Crosses between *sbds^{+/−}* zebrafish with two copies of the transgene. **(B)** Neutrophil count in 5 dpf larvae.

Ebert, 2010; Bursac et al, 2014). To explore the role of Tp53 in our models, we outcrossed the *sbds^{−/−}* with the *tp53^{M214K}* zebrafish mutant (Berghmans et al, 2005). We created *sbds^{+/−};tp53^{M214K/M214K}* zebrafish and incrossed them (Fig 7A). The homozygous *tp53^{M214K/M214K}* background did not rescue neutropenia in *sbds^{−/−}* fish at 10 dpf (Fig 7B). Because our previous results showed a significant increase in *cdkn1a* levels, we next determined whether this was dependent on Tp53 activity. *tp53* and *cdkn1a* levels in the *sbds^{−/−}* mutants with *tp53^{M214K/M214K}* background were not significantly different from their WT siblings (*sbds^{+/+}; tp53^{M214K/M214K}*) (Fig 7C).

To further investigate the role of Tp53, we outcrossed our transgenic line with the *tp53^{M214K}* mutant (Fig 7D). When we incrossed the 1x *Tg(SBDS^{R126T});sbds^{−/−};tp53^{M214K/M214K}* we saw similar results in development and survival (Fig 7E) as previously observed in the tp53 WT background (Fig 5B). We found that *eif6* and the ribosomal protein mRNA levels were not affected in any of the genotypes (Fig 7F). We also observed a significant decrease of *cdkn1a* and *mdm2* levels in all *tp53^{M214K/M214K}* backgrounds compared with the WT (*sbds^{+/+};tp53^{+/+}*). Interestingly, all lipid metabolism markers analyzed were significantly decreased as previously seen in the tp53 WT background (Figs 5D and 7F).

## Discussion

Defects in ribosomal biogenesis profoundly affect the development with variable phenotypes. The ribosomopathy SDS results in neutropenia, pancreatic exocrine insufficiency, and skeletal anomalies. SDS occurs almost exclusively from biallelic mutations in *SBDS*, which encodes a ribosomal assembly factor that is highly conserved from archaea to yeast to vertebrates. We recently reported that the *sbds*-null zebrafish phenocopies the human disease but was lethal during the larval stage (10–21 dpf). Here, we report a new zebrafish transgenic line expressing the mutation SBDS^{R126T} present in a few SDS patients. Unlike the mouse strain that dies before birth (Tourlakis et al, 2012), the zebrafish *sbds* KO lines expressing the disease-associated *SBDS^{R126T}* variant survive to adulthood and are fertile. These fish displayed a significantly lower number of neutrophils compared with the WT in early development (10 dpf), but they recovered to normal levels in adult fish, as can be observed in patients with SDS.

Differences in survival and phenotypes between the zebrafish model expressing one or two copies of the transgene may be because of differences in the expression quantity of Sbds. Interestingly, similar results were found in the three different SDS mouse models. (1) In the Sbds KO, Sbds was important during embryonic development. At E3.5, *Sbds^{−/−}* mice showed that total ablation of Sbds did not affect development before implantation. However, *Sbds^{−/−}* embryos displayed severe growth, structural defects and failed to develop before E6.5 (Zhang et al, 2006a). (2) The knock-in strains *Sbds^{R126T/−}* and *Sbds^{R126T/R126T}* survived longer than the Sbds KO but displayed severe growth impairment and none lived beyond birth. Interestingly, *Sbds^{R126T/−}* embryos were significantly smaller than *Sbds^{R126T/R126T}* embryos at E18.5 (Tourlakis et al, 2015). (3) Pancreas-specific Sbds KO mice were created using the cre recombinase under the pancreatic transcription factor 1a promoter (Ptf1a-cre). In this organismal model, the *Sbds*-null mouse pancreas (*Sbds^{P−/−}*) showed acinar cell hypoplasia at E18.5, but (*Sbds^{P−/R126T}*) was similar to the controls at that same age (Tourlakis et al, 2012). The pancreas in the *Sbds^{P−/R126T}* background suggested activation of tp53 pathway by increased Tp53 protein levels and *cdkn1a* mRNA up-regulation at 15 d of age. Defects in the pancreas were tp53-dependent because complete genetic ablation of tp53 alleviated the phenotype. However, absence of tp53 did not alleviate the lethality and growth impairment in the constitutive SDS mouse embryo. These results showed Tp53-dependent and Tp53-independent mechanism in the syndrome's pathophysiology.

The Sbds^{R126T} mouse results are from a knock-in mutation, however, little is known about the transcriptional and developmental regulation of the *Sbds* promoter. The zebrafish model that we generated is a transgenic line expressing the SBDS^{R126T} via the *ubi* promoter which drives constitutive transgene expression during all developmental stages and adult organs (Mosimann et al, 2011). The tissues most affected by complete loss of Sbds may be relieved in a dose-dependent fashion with SBDS^{R126T}.

We observed an accumulation of Eif6 in our SDS zebrafish models. EIF6 plays an essential role in ribosome maturation and translation. It acts as an anti-association factor to hold nascent pre-60S and mature post-termination 60S in a translationally inactive state (Jaako et al, 2022). As previously described (Finch et al, 2011; Tourlakis et al, 2015), our results confirmed that SBDS^{R126T} is functional enough to release the Eif6 from the 60S. When there are

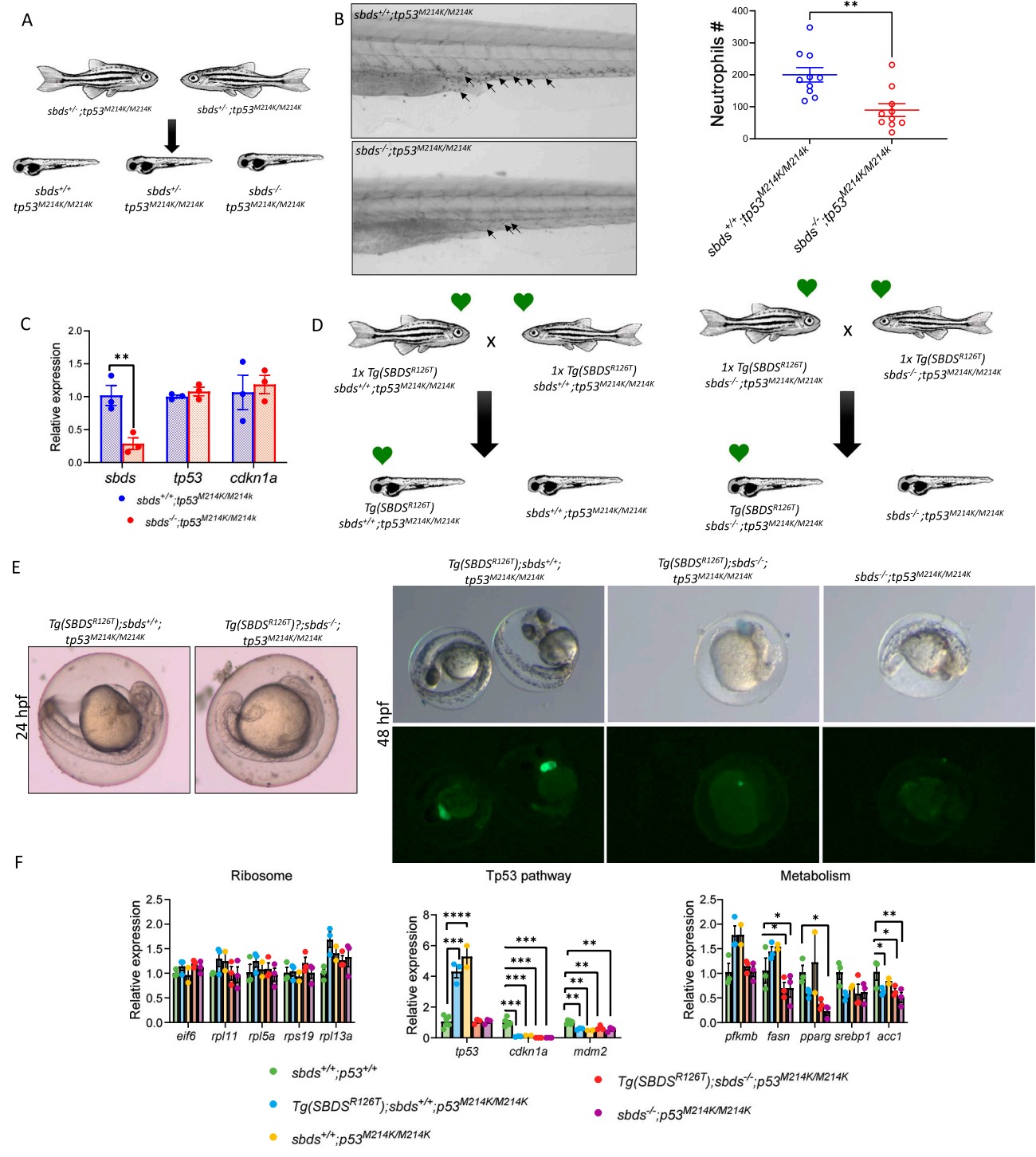

**Figure 7. Tp53$^{M214K}$ does not rescue neutropenia or survival in *sbds* mutants.**
**(A)** Scheme of an incross of sbds$^{+/-}$; *p53$^{M214K/M214K}$*. **(B)** Sudan black staining and neutrophil count in 10 dpf larvae. **(C)** qRT-PCR analysis of *sbds*, *tp53*, and *cdkn1a* in the *p53$^{M214K/M214K}$* background comparing *sbds* KO and WT siblings. **(D)** Incross between WT sbds fish with 1x copy of the transgenic line in the *tp53$^{M214K/M214K}$* background. **(E)** Embryos obtained from crosses at 24 and 48 hpf. **(F)** qRT-PCR analysis of critical genes involved in ribosomes, metabolism, and tp53 pathways.

high levels of SBDS$^{R126T}$ in the cell (two copies of the transgene in the *sbds* KO background), there may be more functional 60S and some 60S-Eif6 (inactive form) that may accumulate in the cytoplasm leading to a partial tp53 pathway activation. This small

amount of Eif6 still bound to the large subunit may form the extra peak observed in the polysome profile (Fig 2C). However, when there is low expression of the SBDS$^{R126T}$ (one copy of the transgene in the *sbds* KO background) there is more 60S-Eif6 than functional

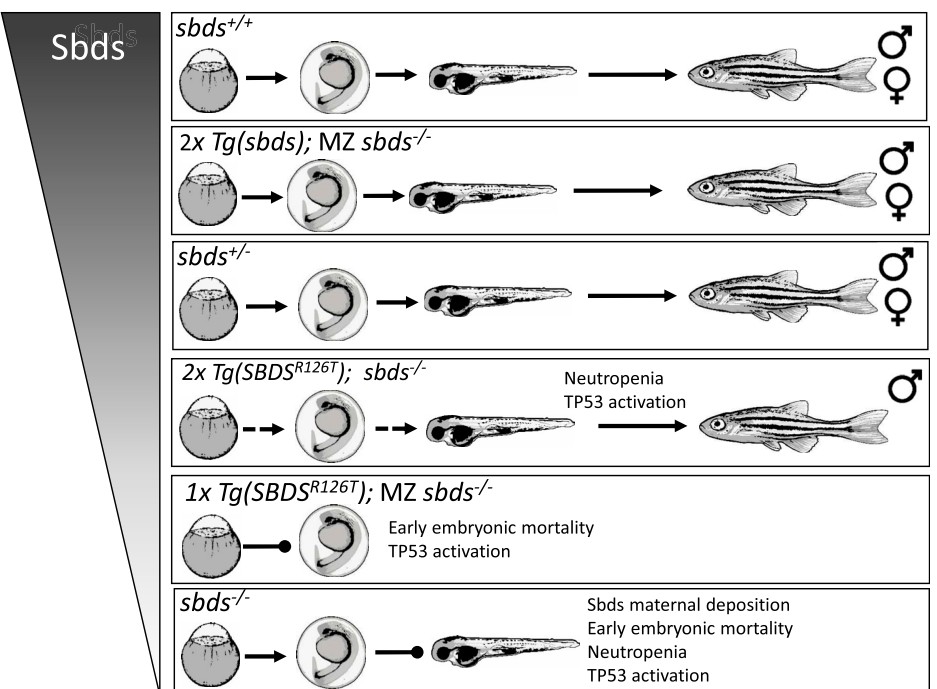

**Figure 8. Summary of variable SBDS protein levels on the sbds-null zebrafish.**
Sbds dose affects survival during development. Note the difference in zebrafish WT sbds rescue *Tg(ubi:sbds:pA)* and the transgenic line expressing the human *SBDS^{R126T} Tg(ubi:SBDS^{R126T}: pA)*. Closed arrows show mortality and dash arrows shows partial mortality. MZ, maternal zygotic.

60S, leading to cell death, in a tp53-independent way. In patients with SDS, somatic *EIF6* mutations provide a compensatory mechanism that is protective against disease transformation to leukemia (Kennedy et al, 2021; Tan et al, 2021).

Tp53 mutation M214K, did not rescue neutropenia or survival in our *sbds* mutant zebrafish models. In addition, the *sbds* knockdown zebrafish model showed that loss of Tp53 did not rescue the developmental abnormalities or its early mortality (Provost et al, 2012). In SDS patients, acquisition of TP53 biallelic mutations conferred a higher risk to develop acute myeloid leukemia (Kennedy et al, 2021). Despite the zebrafish models not developing leukemia, the Tp53 activation may be important in the cellular fitness long-term.

Activation of *tp53/cdkn1a* persisted throughout life but did not have any affect survival. Interestingly, the level of expression of the SBDS^{R126T} in the *sbds*-null background resulted in increased defective embryos. Those fish which survived development also displayed a suppression of female sex differentiation (Fig 8). Genetic and environmental factors influence sex determination in zebrafish (von Hofsten & Olsson, 2005; Liew & Orbán, 2014). In particular, tp53 activation can favor the determination of male gonad transformation and differentiation in zebrafish (Rodríguez-Marí et al, 2010; Rodriguez-Mari & Postlethwait, 2011). In a zebrafish model using *fancl* mutants, Rodríguez-Marí et al (2010) observed that the tp53 activated the apoptotic processes in the male gonads resulting in the removal of oocytes and tilted the balance toward testicular differentiation. Tp53 activation affects sex tissue-specific apoptosis that tilts the hormonal balance to the formation of testis (Rodríguez-Marí et al, 2010).

In conclusion, our new zebrafish transgenic line expressing the SBDS^{R126T} demonstrated that the amount of mutant SBDS protein is important for zebrafish development and activates the *tp53/cdkn1a* pathway. However, inactivation of tp53 does not rescue neutropenia or survival of *sbds* mutants. Other pathways may be implicated in the SDS pathophysiology, such as lipid metabolism, that need further study.

# Materials and Methods

### Zebrafish husbandry

Animals were maintained according to standard protocols at 28.5°C. Zebrafish strains AB, *tp53^{M214K}*, *sbds^{nu132}* (here denoted as *sbds^-*) were raised in a circulating aquarium system (AQUA) at 28.5°C in a 14/10 h light/dark cycle and maintained according to standard protocols.

### Transgenic lines

To create the *SBDS^{R126T}* transgenic line, we amplified human *SBDS* from the plasmid GE Dharmacon MHS6278 using Q5 High-Fidelity DNA Polymerase (NEB) and subcloned the cDNA into the pDONR221 plasmid (Invitrogen, Thermo Fisher Scientific) by BP reaction. An inverse PCR reaction was carried out to introduce the corresponding point mutation R126T. We used the 5'E: ubi and 3' pE-GFP using the gateway method (Mosimann et al, 2011). Finally, a recombination reaction was carried out using these three plasmids and pDestTol2CG, that includes the *cmlc2*: EGFP-pA expression cassette, (*cmlc2* cardiac myosin light chain 2a promoter that drives

GFP in the hearts and allows to screen the presence of the transgene from 2 dpf) to create a final construct *ubi:SBDS^R126T:pA*.

## Polysome profile in livers

We collected 1-yr-old male fish livers of the five different genotypes (*sbds^+/+*, *sbds^+/−*, *Tg(ubi:SBDS^R126T:pA);sbds^+/+*, *Tg(ubi:SBDS^R126T:pA); sbds^+/−* *Tg(ubi:SBDS^R126T:pA);sbds^−/−*) for polysome profiling. First, we treated the fish with 100 $\mu$g/ml cycloheximide for 10 min in fish water followed by euthanasia by rapid chilling. Dissections of the livers were performed under the stereoscope as previously described by Gupta & Mullins (2010), and flash-frozen at −80°C. Livers were thawed and lysed at 4°C in polysome lysis buffer (10 mM Tris–HCl pH 7.4, 5 mM MgCl$_2$, 100 mM KCl, 1% Triton X-100, 2 mM DTT, 200 $\mu$g/ml cycloheximide, complete EDTA-free protease inhibitor cocktail [Roche], and 100 U/ml SUPERaseIn [Invitrogen]) by repeated trituration through a #26 gauge needle and incubated on ice for 10 min. The lysate was cleared by centrifugation at 1,000$g$ for 10 min at 4°C. The supernatant was carefully layered onto a 12 ml 10–50% sucrose gradient made in 10 mM Tris–HCl pH 7.4, 5 mM MgCl$_2$, 100 mM KCl, 2 mM DTT, and 100 $\mu$g/ml cycloheximide. Lysates were ultracentrifuged at 210,000$g$ using an SW-41 Ti rotor (331362; Beckman Coulter) at 4°C for 2.5 h. Gradients were analyzed using a BioComp Poston Gradient Fractionator with monitoring of absorbance at 254 nm.

## qRT–PCR

Brain, liver, and intestinal tissues were isolated from 1-yr-old zebrafish. The 10 dpf larvae from crosses between *Tg(ubi:SBDS^R126T:pA);sbds^+/−* and *sbds^+/−* were fin-clipped, genotyped, and the body was kept in individual PCR tubes. Pools of 3–7 larvae were used for RNA extraction. We used at least three biological replicates for each experiment. All genotypes were from the same clutch. Larvae pools/tissues were homogenized, and RNA was isolated using TRIzol (Invitrogen, ThermoFisher Scientific). cDNA was synthesized using iScript (Bio-Rad). Primer sequences are available from the corresponding author by request. mRNA expression in mutants relative to WT was normalized to *β-actin* and calculated according to the ΔΔCT method.

## Western blotting

Larvae and fins were collected and boiled for 10 min in Laemmli buffer (Bio-Rad) with 2-mercaptoethanol. The following antibodies were purchased to detect SBDS (sc-271600; Santa Cruz Biotechnology), Tp53 (55342; Anaspec), EIF6 (NBP2-16975; NovusBio), RPL11 (18163; Cell Signaling Technology), RPL5 (14568; Cell Signaling Technology), *β*-tubulin (2146; Cell Signaling Technology), and *β*-actin (sc-47778; Santa Cruz Biotechnology).

## Histology

Fish were euthanized and fixed in 4% PFA and decalcified using 5% trichloroacetic acid. Sectioning and H&E staining was carried out in the Cleveland Clinic Histology Core.

## Cell preparation and flow cytometry

Zebrafish kidneys were isolated under the stereoscope and mechanically dissociated using a 40-$\mu$m filter, rinsed in PBS-4%FBS, and centrifuged for 5 min at 800$g$ twice. The supernatant was aspirated, and the cells were resuspended in PBS-4% FBS.

## Neutrophil detection

We fixed the 10 dpf larvae for 2 h at RT, rinsed three times with PBS for 10 min, and added Sudan black (380B-1KT; Sigma-Aldrich) for 20 min, then rinsed twice in 70% EtOH for 2 min and in PBS three times. The larvae were bleached for depigmentation and imaging.

## Imaging

All images were taken using ZEISS stereoscopes (Stemi 508 and Discovery V8) and an AxioImager M2 microscope with a camera (Axiocam). Image analysis was carried out using ImageJ.

## Statistics

Descriptive and analytical statistics were performed with Prism 6.0 (GraphPad Software). Parametric data are presented as mean ± SEM. The n values are indicated by dots in histograms; each individual n value represents a different animal. Statistical analysis used unpaired two-tailed $t$ tests or one-way ANOVA with Tukey's multiple-comparisons test. $P < 0.05$ was used to indicate a significant difference.

## Study approval

All zebrafish experiments were approved by the Animal Care Usage Committee at the Cleveland Clinic.

# Supplementary Information

# Acknowledgements

This work was supported by grants from VeloSano, NIH R01 HL128173, NIH R21 CA159203, NIH R01 DK132812, and Lisa Dean Moseley Foundation (SJ Corey), VeloSano (U Oyarbide), and R35GM142634 from NIGMS (E Calo). We thank M Fleece for his contribution with the scientific drawings.

## Author Contributions

U Oyarbide: conceptualization, resources, data curation, formal analysis, funding acquisition, validation, investigation, visualization, methodology, project administration, and writing—original draft, review, and editing.
AN Shah: data curation, formal analysis, investigation, methodology, and writing—review and editing.

M Staton: data curation and methodology.

M Snyderman: data curation and methodology.

A Sapra: data curation and methodology.

E Calo: formal analysis, supervision, investigation, methodology, and writing—original draft, review, and editing.

SJ Corey: conceptualization, resources, supervision, funding acquisition, project administration, and writing—original draft, review, and editing.

## Conflict of Interest Statement

The authors declare that they have no conflict of interest.

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
