## [Reviewer comments · Life Science Alliance]

Life Science Alliance

SBDS R126T rescues survival of sbds^{-/-} zebrafish in a dose-dependent manner independently of Tp53

Usua Oyarbide, Arish Shah, Morgan Staton, Matthew Snyderman, Adya Sapra, Eliezer Calo, and Seth Corey DOI: <https://doi.org/10.26508/lsa.202201856>

Corresponding author(s): *Usua Oyarbide, Cleveland Clinic*

Review Timeline:

Submission Date:	2022-11-25
Editorial Decision:	2022-12-27
Revision Received:	2023-08-02
Editorial Decision:	2023-09-13
Revision Received:	2023-09-26
Accepted:	2023-09-27

Scientific Editor: *Eric Sawey, PhD*

Transaction Report:

December 27, 2022

Re: Life Science Alliance manuscript #LSA-2022-01856-T

Usua Oyarbide
Cleveland Clinic

Dear Dr. Oyarbide,

Thank you for submitting your manuscript entitled "SBDSR126T restores survival of the sbds-null zebrafish model of Shwachman-Diamond syndrome in a dose-dependent manner without abrogating activation of tp53/cdkn1a" to Life Science Alliance. The manuscript was assessed by expert reviewers, whose comments are appended to this letter. We invite you to submit a revised manuscript addressing the Reviewer comments.

Thank you for this interesting contribution to Life Science Alliance. We are looking forward to receiving your revised manuscript.

Sincerely,

B. MANUSCRIPT ORGANIZATION AND FORMATTING:

Reviewer #1 (Comments to the Authors (Required)):

In this study, the authors set out to determine whether transgenic expression of the human SBDS protein carrying a missense SBDS R126T mutation genetically complements sbds null zebrafish. Based on their studies, the authors claim that the R126T mutation is hypomorphic and that the ability to rescue the development of sbds null fish is dose dependent. The sbds null fish carrying the SBDS R126T transgene exhibit neutropenia and suppression of female sex differentiation, which the authors suggest is a consequence of tp53 activation. Unfortunately, I have some major concerns with the work as it is currently presented.

The novelty of the work is unclear based on previously published work on the SBDS R126T mutant allele in mice/cell lines/in vitro (Zhang et al 2006; Turlakis et al 2015; Calamita 2017; Finch et al 2011) showing that this is indeed a hypomorphic allele. The current study does not seem to significantly advance understanding of how the R126T missense mutation impairs SBDS function.

I also have concerns that a key control experiment seems to be lacking. I may have misinterpreted it due to lack of clarity in the labeling/annotation of the mutant proteins/genes, but it is essential to test whether the WT human SBDS protein fully complements the sbds null zebrafish mutant to support the claim that the "SBDS R126T mutant is hypomorphic". For me, the conclusion "Our results show that SBDS R126T is hypomorphic" is not currently supported by the data. However, this aside, we already know from previous studies and from the human genetics that the SBDS R126T missense mutation is indeed hypomorphic and impairs SBDS function compared to WT.

To validate the conclusion that tp53 activation is causing the neutropenia and suppression of sex differentiation in the SBDS R126T fish, it is necessary to remove tp53 genetically.

Additional specific points

1. No page or line numbers makes it difficult to comment on text
2. "Most mutations result in a premature termination codon". This is not true-the donor splice site mutation results in low level expression of the full-length protein.
3. In the discussion: "Most mutations result in a premature termination codon and hypomorphic protein"-the term hypomorphic indicates impaired gene function
4. In the abstract: "However, its higher expression"
Higher expression of what? Please clarify
5. In the first paragraph, it is difficult to understand exactly what has been done. Initially WT zebrafish sbds is expressed in the sbds null background, but then I understand that the authors have expressed the human SBDS R126T mutant. Is that correct? This needs to be stated explicitly. This begs the question whether human WT SBDS can fully rescue the sbds null mutant zebrafish? Clarity of the text is hampered by inconsistent use/explanation of abbreviations used in labeling the figures and as written in the text:
Eg hSBDS/hSBDSR126T/SBDSR126T/sbds/R126T/sbdsnu132/-132/Mut/WT_020
Please keep consistent and explain the nomenclature.
6. ubi promoter: it would be helpful if the text in the discussion is moved to the main results section to provide more explanation for the general reader on the nature of the ubi promoter.
"via the ubi promoter which drives constitutive transgene expression during all developmental stages and adult organs. (Mosimann et al. 2011) "
7. Figure 1A: what is cmlc2? Please explain
8. Figure 2A: why was the time point of 1 yr chosen?

9. Figure 2D: what do the labels indicate-132/Mut/WT_020. Difficult to interpret the data if the labeling is unexplained.
10. "This extra peak was absent in the WT liver calibrations (data not shown)". What is the interpretation of this additional peak?
11. Fig 2G: Eif6 level is increased in 1/3 sbds WT lines expressing human SBDS R126T-please comment on this.
12. Fig 2I, 3D: are the expression levels of the cdkn1a protein altered in the different genotypes?
13. Fig 3C: "Sudan black staining showed a decreased in the number of neutrophils in sbds KO background fish, independently of the presence of SBDS R126T". Please show primary data.
14. "our data show that activation of tp53 pathways involving cdkn1a may account for the neutropenia"
Does removal of tp53 rescue the neutropenia phenotype?
15. SBDS physically interacts with the GTPase Elongation Factor Like 1 (EFL1) to release the Eukaryotic Initiating Factor 6 (EIF6) from the cytoplasmic pre-60S ribosomal subunit. This release facilitates the assembly of the mature 80S ribosome. Reference should be included
16. "These patients present with severe hematological phenotypes.(Zhang et al. 2006b)"
What is the evidence for this statement? It is not provided in the cited reference.

Reviewer #2 (Comments to the Authors (Required)):

This study explores the role of co-occurring mutations in disease pathophysiology. This manuscript has established a zebrafish transgenic model system to investigate the effects of a missense mutation SBDSR126T in a ribosomopathy, Shwachman-Diamond syndrome (SDS). This study is conceptually like the one performed by Zhang S et al, 2006a. However, embryonic lethality was a huge limitation in the Zhang S et al study. To overcome this limitation, the authors generated a transgenic line introducing a transgene carrying the SBDSR126T mutation in a sbds null background. This methodology allowed the authors to manipulate the levels of SBDSR126T in the sbds null mutants and establish that higher levels of SBDSR126T are critical for organism survival and development when present in a sbds null background. The authors perform several techniques, such as protein estimation by western blotting, qPCR for mRNA quantification, sudan black staining for neutrophil counts, and polysome profiling, to come to their conclusion. However, consistency across the article, an adequate explanation of results, and discussion could be much better. Please refer to the following points.

1. Fig 1A - The Schematic could be prepared better to explain the study design. What do the black rectangles flanking Zf-sbds mean?
2. Fig 1B - Please label the sizes of the proteins in all the western blots.
3. Fig 1C - The authors are requested to perform southern blotting to determine the number of copies of the inserted transgene. Using a transgene-independent technique to assess copy numbers will be critical, as transgenes may get silenced after the F1 generation. It is more important for this study as it involved 1x and 2x copy of transgenes to achieve the conclusion of the study.
4. Introduction, Paragraph 3, line 8 - Please rewrite "SbdsR126T/R126 littermates" to "SbdsR126T/R126T littermates".
5. Results, Paragraph 3, line 2 - Please rewrite "Tg(ubi:SBDSR126T;pA) sbds+/-" to "Tg(ubi:SBDSR126T;pA);sbds+/-".
6. Fig 2B - Please rewrite figure legend "(B) Survival rates in one yr-old fish" to "(B) Standard length in one yr-old fish".
7. Fig 2D - In the flow cytometry analysis bar plots, samples representing sbds-/- (Green dots) are plotted for erythrocytes, lymphocytes, myeloid, and precursors. According to the authors, the sbds-/- mutants die between 10-21dpf. How did the authors perform these analyses at 1 year for the sbds-/- animals? Is this a case of mislabelling? sbds+/- samples are not plotted in the "no R126T" bars. Please provide clarification.
8. Fig 2E - Provide clear and large labeling (inset) for the polysome profiles. What does "+/132" stand for? Please explain the genotypes in the figure legend.
9. Results, Paragraph 4, line 3 - The authors refer to 'mutant' in the text. Please refer to sbds+/- mutants for clarity.
10. Results, Paragraph 4, line 9 - The authors state, "This extra peak was absent in the WT liver calibrations (data not shown)." In fig 2F, they provide the polysome profile for the WT liver (in red).
11. Figure 2I - The authors state that they observed a decrease in rpl11 and rpl5 protein levels and accumulation of Eif6 protein in 10dpf sbds-/- larvae from their previous study. In this study, the authors evaluated the levels of these proteins by western blotting of fin lysates from the transgenic adult fish (1 yr-old) and found the ribosomal proteins to be similar in WT, Tg(hSBDSR126T)sbds+/+ and Tg(hSBDSR126T)sbds-/. However, the authors can detect the accumulation of Eif6 in the fin lysates of Tg(hSBDSR126T)sbds+/+ and Tg(hSBDSR126T)sbds-/. Have the authors tested the expression of these proteins from 1-year-old liver lysates of WT, Tg(hSBDSR126T)sbds+/+, and Tg(hSBDSR126T)sbds-/-? Since the liver is more metabolically active than the fin, it would be interesting to investigate the ribosomal pathway protein levels from liver lysates.
12. Results, Paragraph 6, line 8 - Please rewrite "showed a decreased" to "showed a decrease".
13. Figure 4A - The authors indicate healthy embryo (black star) from Tg(ubi:SBDSR126T;pA);sbds-/. However, this embryo visually looks developmentally delayed. In several instances, the authors demonstrate the activation of p53 and stress response pathways. It will be critical to analyze the apoptosis and cell proliferation in the various transgenic animals in sbds-/- null

background to validate the physiological impact of activating the aforesaid pathways.

14. Figure 4C - The authors write in the text that they "checked the mRNA levels of the three critical glycolysis enzymes (gck, pfkmb and pyrk)"; however, the expression levels of only pfkmb are displayed in the graphs.
15. Figure 4G - Please use X-axis split for the standard length (SL) bar plot. This will help to visualize and appreciate the data in a better way.
16. Figure 5A - How do the authors distinguish between Tg(ubi:SBDSR126T;pA);sbds^{-/-} embryos (arising from the cross male Tg(ubi:SBDSR126T;pA);sbds^{-/-} and female Tg(ubi:SBDSR126T;pA);sbds^{-/-}) having 1 transgene and 2 transgenes?
17. Figure 5C - Does the graph presented for the percentage of deformed embryos correspond to the sbds^{+/+} and sbds^{-/-} only? Do these embryos have the transgene? If these embryos have transgene, then according to the breeding cross, 25% of the embryos must have 2 copies of transgene in the sbds^{-/-} MZ background and would be able to survive. In that case, the statement by the authors "Interestingly, embryonic development was defective in 25% of the sbds^{-/-} embryos after 1 dpf, and the mortality was 100% after 3 dpf (Figure 5B-C), presence of transgene did not ameliorate the defects" does not make sense.
18. Figure 5D - qPCR analysis corresponding to sbds gene is confusing. The bar graphs suggest an increase in sbds mRNA levels in TgR126T;sbds^{+/+} and TgR126T;sbds^{-/-} as compared to sbds^{+/+} 2 dpf embryos. Does this mean the TgR126T transgene triggers endogenous sbds expression?
19. Figure 5D - qPCR analysis corresponding to hSBDS mRNA seems incorrect. There are data points visible for sbds^{-/-}. How can you quantitate an mRNA derived from the transgene which is not present in the sample? Similarly, why are the mRNA levels corresponding to hSBDS lower in TgR126T;sbds^{-/-} as compared to TgR126T;sbds^{+/+}?
20. Figure 6: The authors demonstrate that high expression (2 transgene) of SBDSR126T rescues neutropenia at 5dpf. However, at 10dpf, the high expression of SBDSR126T is unable to rescue neutropenia. The authors need to talk about this in the discussion section.
21. Results, Paragraph 9, line 6 - Please rewrite "cas9" to "casp9".
22. Discussion, Paragraph 6, line 3 - Please rewrite "SBDS protein is important for zebrafish development" to "SBDS protein is important for zebrafish development".
23. Does the SBDSR126T allele in the SDS patients display any sex specificity (prevalent in males)?
24. The authors need to emphasize the use of maternal zygotic sbds null mutants in their results.
25. Does suppression of cdkn1a in the sbds null background result in reversed neutropenia?
26. Performing qRT-PCR for all the transgenics (sbds^{+/+}, sbds^{-/-}, TgR126T;sbds^{+/+}, TgR126T;sbds^{-/-} 1copy and 2 copy) at Day1 , Day2 , Day5 and Day10 of development would be informative.

1st August 2023

Thank you for giving us the opportunity to submit a revised draft of our manuscript titled "SBDS^{R126T} rescues survival of *sbds*^{-/-} zebrafish in a dose dependent manner independently of Tp53" to Life Science Alliance. We appreciate the time and effort that you and the reviewers have dedicated to providing your valuable feedback on our manuscript. We are grateful to the reviewers for their insightful comments on our paper. We have been able to incorporate changes to reflect most of the suggestions provided by the reviewers. All the changes within the manuscript are denoted in **red font**.

Here is a point-by-point response to the reviewers' comments and concerns:

Reviewer 1

Comment 1: The novelty of the work is unclear based on previously published work on the SBDS R126T mutant allele in mice/cell lines/in vitro (Zhang et al 2006; Turlakis et al 2015; Calamita 2017; Finch et al 2011) showing that this is indeed a hypomorphic allele. The current study does not seem to significantly advance understanding of how the R126T missense mutation impairs SBDS function.

Response: Arguably, the most important question in the biology of bone marrow failure disorders is what constitutes the pathogenic mechanism(s). The current dogma, mostly established from another different marrow failure ribosomopathy (Diamond-Blackfan anemia) is the TP53 pathway. Here, we demonstrated that the tp53 pathway is activated in both the zebrafish KO and the transgenic line expressing R126T. But, Tp53 inactivation does not rescue neutropenia or survival – we consider this novel and extremely important to the field. This leads to the hypothesis that there are other pathways that may affect SDS pathophysiology and suggests that not all ribosomopathies behave the same. As importantly, we have generated a novel model organism which did not receive any wild-type mRNA or protein because maternal eggs were derived from parental transgenics.

Comment 2: I also have concerns that a key control experiment seems to be lacking. I may have misinterpreted it due to lack of clarity in the labeling/annotation of the mutant proteins/genes, but it is essential to test whether the WT human SBDS protein fully complements the *sbds* null zebrafish mutant to support the claim that the "SBDS R126T mutant is hypomorphic". For me, the conclusion "Our results show that SBDS R126T is hypomorphic" is not currently supported by the data. However, this aside, we already know from previous studies and from the human genetics that the SBDS R126T missense mutation is indeed hypomorphic and impairs SBDS function compared to WT.

Response: We agree that SBDS^{R126T} has been previously determined to be hypomorphic. However, in our study we demonstrated that the amount of SBDS is important for the embryonic development. We revised the text: As previously described, our results confirm that SBDS^{R126T} is hypomorphic, and it is functional enough to release the EIF6 from the 60S.

Comment 3: To validate the conclusion that tp53 activation is causing the neutropenia and suppression of sex differentiation in the SBDS R126T fish, it is necessary to remove tp53 genetically.

*Response: We outcrossed the *sbds* lines with the tp53 M214K. Our studies demonstrated that tp53^{M214K} did not rescue neutropenia or survival. Moreover, we also demonstrated that *cdkn1a* is dependent of tp53 presence. (Figure 7)*

No page or line numbers makes it difficult to comment on text.

Response: We added page and line numbers

Comment 4: Most mutations result in a premature termination codon". This is not true-the donor splice site mutation results in low level expression of the full-length protein.

Response: We revised the sentence and added this statement: Most commonly, SBDS mutations are found in exon 2 and lead to disruption of a donor splice and a frameshift mutation (C83fs) or protein truncation after introduction of a stop codon (K62X). C83fs mutation produces reduced expression of full-length protein (Austin et al. 2005). While a few patients are homozygous for the splice donor mutation, homozygous mutants for K62X have not been described, suggesting that complete loss of SBDS is lethal (Boocock et al. 2003; Austin et al. 2005; Shamma et al. 2005).

Comment 5: In the discussion: "Most mutations result in a premature termination codon and hypomorphic protein"-the term hypomorphic indicates impaired gene function.

Response: We deleted this comment.

Comment 6: In the abstract: "However, its higher expression" Higher expression of what? Please clarify.

Response: We rewrote the sentence for greater clarity: "We showed that one copy of the SBDS^{R126T} transgene permitted the establishment of maternal sbds-null fish which produced defective embryos with cdkn1a upregulation. However, two copies of the transgene permitted full development and lifespan.

Comment 7: In the first paragraph, it is difficult to understand exactly what has been done. Initially WT zebrafish sbds is expressed in the sbds null background, but then I understand that the authors have expressed the human SBDS R126T mutant. Is that correct? This needs to be stated explicitly. This begs the question whether human WT SBDS can fully rescue the sbds null mutant zebrafish? Clarity of the text is hampered by inconsistent use/explanation of abbreviations used in labeling the figures and as written in the text:

e.g., hSBDS/hSBDSR126T/SBDSR126T/sbds/R126T/sbdsnu132/-132/Mut/WT_020

Please keep consistent and explain the nomenclature.

Response: We have revised the whole manuscript and change the nomenclature accordingly: sbds^{+/+}, sbds^{+/-}, sbds^{-/-}, Tg(SBDS^{R126T})

Comment 8: ubi promoter: it would be helpful if the text in the discussion is moved to the main results section to provide more explanation for the general reader on the nature of the ubi promoter. "via the ubi promoter which drives constitutive transgene expression during all developmental stages and adult organs.(Mosimann et al. 2011) "

Response: We added this sentence in the methods

Comment 9: Figure 1A: what is cmlc2? Please explain.

Response: cmlc2 stands for cardiac myosin light chain 2. The rationale for its use was added in the methods section 312-313, and we added it to Figure 1A legend too.

Comment 10: Figure 2A: why was the time point of 1 yr chosen?

Response: This is a midlife in the lifespan of zebrafish.

Comment 11: Figure 2D: what do the labels indicate-132/Mut/WT_020. Difficult to interpret the data if the labeling is unexplained.

Response: We changed this legend

Comment 12: "This extra peak was absent in the WT liver calibrations (data not shown)". What is the interpretation of this additional peak?

Response: we deleted that comment

Comment 13: Fig 2G: Eif6 level is increased in 1/3 sbds WT lines expressing human SBDS R126T-please comment on this.

Response: This may be due to variability among fish. This individual fish also expressed higher levels of sbds, rpl5, rpl11, and tp53 (compared to a-tubulin)

Comment 14: Fig 2I, 3D: are the expression levels of the cdkn1a protein altered in the different genotypes?

Response: Unfortunately, we have tried six commercially available antibodies, but none cross-react with the zebrafish p21.

Comment 15: Fig 3C: "Sudan black staining showed a decreased in the number of neutrophils in sbds KO background fish, independently of the presence of SBDS R126T". Please show primary data.

Response: We added the figure.

Comment 16: "our data show that activation of tp53 pathways involving cdkn1a may account for the neutropenia" Does removal of tp53 rescue the neutropenia phenotype?

Response: We revised this statement. Our new results showed that tp53 M214K mutant does not rescue neutropenia in the sbds KO background (see new Figure 7)

Comment 17: SBDS physically interacts with the GTPase Elongation Factor Like 1 (EFL1) to release the Eukaryotic Initiating Factor 6 (EIF6) from the cytoplasmic pre-60S ribosomal subunit. This release facilitates the assembly of the mature 80S ribosome. Reference should be included.

Response: We have added this important reference (Finch et al 2011. Genes Dev).

Comment 18: These patients present with severe hematological phenotypes.(Zhang et al. 2006b)" What is the evidence for this statement? It is not provided in the cited reference.

Response: We deleted this statement.

Reviewer 2

Comment 1: Fig 1A - The Schematic could be prepared better to explain the study design. What do the black rectangles flanking Zf-sbds mean?

Response: We clarified that figure.

Comment 2: Fig 1B - Please label the sizes of the proteins in all the western blots.

Response: Done

Comment 3: Fig 1C - The authors are requested to perform southern blotting to determine the number of copies of the inserted transgene. Using a transgene-independent technique to assess copy numbers will be critical, as transgenes may get silenced after the F1 generation. It is more important for this study as it involved 1x and 2x copy of transgenes to achieve the conclusion of the study.

Response: We measured mRNA levels of the human SBDS as shown in Figure 1D, we used as negative controls the non-green heart siblings, the Ct values for these negative controls were

undetermined or higher than 33 (figure 1D). We also analyzed protein levels in 1x and 2x copies of the transgene (Figure 1E).

Comment 4: Introduction, Paragraph 3, line 8 - Please rewrite "SbdsR126T/R126 littermates" to "SbdsR126T/R126T littermates".

Response: Done

Comment 5: Results, Paragraph 3, line 2 - Please rewrite "Tg(ubi:SBDSR126T:pA) sbds+/-" to "Tg(ubi:SBDSR126T:pA);sbds+/-".

Response: Done

Comment 6: Fig 2B - Please rewrite figure legend "(B) Survival rates in one yr-old fish" to "(B) Standard length in one yr-old fish".

Response: Done

Comment 7: Fig 2D - In the flow cytometry analysis bar plots, samples representing sbds-/- (Green dots) are plotted for erythrocytes, lymphocytes, myeloid, and precursors. According to the authors, the sbds-/- mutants die between 10-21dpf. How did the authors perform these analyses at 1 year for the sbds-/- animals? Is this a case of mislabelling? sbds+/- samples are not plotted in the "no R126T" bars. Please provide clarification.

Response: We thank the reviewer for catching this error in labeling. The green dots correspond to sbds+/- and the red to sbds-/-.

Comment 8: Fig 2E - Provide clear and large labeling (inset) for the polysome profiles. What does "+/132" stand for? Please explain the genotypes in the figure legend.

Response: Done

Comment 9: Results, Paragraph 4, line 3 - The authors refer to 'mutant' in the text. Please refer to sbds+/- mutants for clarity.

Response: Done

Comment 10: Results, Paragraph 4, line 9 - The authors state, "This extra peak was absent in the WT liver calibrations (data not shown)." In fig 2F, they provide the polysome profile for the WT liver (in red).

Response: We deleted this statement.

Comment 11: Figure 2I - The authors state that they observed a decrease in rpl11 and rpl5 protein levels and accumulation of Eif6 protein in 10dpf sbds-/- larvae from their previous study. In this study, the authors evaluated the levels of these proteins by western blotting of fin lysates from the transgenic adult fish (1 yr-old) and found the ribosomal proteins to be similar in WT, Tg(hSBDSR126T)sbds+/+ and Tg(hSBDSR126T)sbds-/. However, the authors can detect the accumulation of Eif6 in the fin lysates of Tg(hSBDSR126T)sbds+/+ and Tg(hSBDSR126T)sbds-/. Have the authors tested the expression of these proteins from 1-year-old liver lysates of WT, Tg(hSBDSR126T)sbds+/+, and Tg(hSBDSR126T)sbds-/? Since the liver is more metabolically active than the fin, it would be interesting to investigate the ribosomal pathway protein levels from liver lysates.

Response: The reviewer raises an interesting point, which we plan to explore in future studies.

Comment 12: Results, Paragraph 6, line 8 - Please rewrite "showed a decreased" to "showed a decrease".

Response: Done

Comment 13: Figure 4A - The authors indicate healthy embryo (black star) from Tg(ubi:SBDSR126T:pA);sbds^{-/-}. However, this embryo visually looks developmentally delayed. In several instances, the authors demonstrate the activation of p53 and stress response pathways. It will be critical to analyze the apoptosis and cell proliferation in the various transgenic animals in sbds^{-/-} null background to validate the physiological impact of activating the aforesaid pathways.
Response: We agree about the developmental delay and added a statement in the text. Because the sbds-null fish die by 15-21 dpf, we previously reported (Oyarbide, et al, JCI Insights) decrease proliferation in the liver and pancreas, but we were unable to determine apoptosis. We are trying to optimize this staining.

Comment 14: Figure 4C - The authors write in the text that they "checked the mRNA levels of the three critical glycolysis enzymes (gck, pfkmb and pyrkl)"; however, the expression levels of only pfkmb are displayed in the graphs.

Response: We have modified this paragraph: We previously showed that mRNA levels of one important enzyme in the pathway of glycolysis, phosphofructokinase (pfkmb), was downregulated in the sbds^{-/-} fish (Oyarbide et al. 2020). We checked the pfkmb mRNA levels in the Tg(SBDSR126T)sbds^{-/-} and it was decreased in the normal and deformed larvae compared to Tg(SBDSR126T)sbds^{+/+}.

Comment 15: Figure 4G - Please use X-axis split for the standard length (SL) bar plot. This will help to visualize and appreciate the data in a better way.

Response: Done.

Comment 16: Figure 5A - How do the authors distinguish between Tg(ubi:SBDSR126T:pA);sbds^{-/-} embryos (arising from the cross male Tg(ubi:SBDSR126T:pA);sbds^{-/-} and female Tg(ubi:SBDSR126T:pA);sbds^{-/-}) having 1 transgene and 2 transgenes?

Response: We cannot discern 1x v. 2x on the basis of a green heart intensity, and to improve on the signal:noise we pooled multiple larvae.

Comment 17: Figure 5C - Does the graph presented for the percentage of deformed embryos correspond to the sbds^{+/+} and sbds^{-/-} only? Do these embryos have the transgene?

Response: sbds^{-/-} (or sbds^{+/+}) also have SBDS R126T either as a 1x or 2x or none.

Comment 18: If these embryos have transgene, then according to the breeding cross, 25% of the embryos must have 2 copies of transgene in the sbds^{-/-} MZ background and would be able to survive. In that case, the statement by the authors "Interestingly, embryonic development was defective in 25% of the sbds^{-/-} embryos after 1dpf, and the mortality was 100% after 3 dpf (Figure 5B-C), presence of transgene did not ameliorate the defects" does not make sense.

Response: The reviewer makes an interesting observation, which we also recognized but didn't have a clearcut explanation and hence, did not discuss. The F1 all survive with more males than females and with less fecundity among females. Surprisingly, when the F1 breed, the F2 are deformed and die. We hypothesize that there is some sort of serial exhaustion, which may be difficult to identify by single cell RNA-Seq.

Comment 19: Figure 5D - qPCR analysis corresponding to sbds gene is confusing. The bar graphs suggest an increase in sbds mRNA levels in TgR126T;sbds^{+/+} and TgR126T;sbds^{-/-} as compared to sbds^{+/+} 2 dpf embryos. Does this mean the TgR126T transgene triggers endogenous sbds expression?

Response: We clarified the labeling in the legend. No, the TgR126T did not affect transcriptional expression of sbds.

Comment 20: Figure 5D - qPCR analysis corresponding to hSBDS mRNA seems incorrect. There are data points visible for sbds^{-/-}. How can you quantitate an mRNA derived from the transgene which is not present in the sample? Similarly, why are the mRNA levels corresponding to hSBDS lower in TgR126T;sbds^{-/-} as compared to TgR126T;sbds^{+/+}?

Response: We thank the review for catching this and have corrected this.

Comment 21: Figure 6: The authors demonstrate that high expression (2 transgene) of SBDSR126T rescues neutropenia at 5dpf. However, at 10dpf, the high expression of SBDSR126T is unable to rescue neutropenia. The authors need to talk about this in the discussion section.

Response: We have added a brief discussion of these findings to the discussion section. It demonstrates a dosage effect that 1x will not rescue but 2x will.

Comment 22: Results, Paragraph 9, line 6 - Please rewrite "cas9" to "casp9".

Response: Done

Comment 23: Discussion, Paragraph 6, line 3 - Please rewrite "SBDS protein is important for zebrafish development" to "SBDS protein is important for zebrafish development".

Response: Corrected.

Comment 24: Does the SBDSR126T allele in the SDS patients display any sex specificity (prevalent in males)?

Response: We don't have access to clinical information of these rare patients.

Comment 25: The authors need to emphasize the use of maternal zygotic sbds null mutants in their results.

Response: We have emphasized this in the abstract and discussion sections.

Comment 26: Does suppression of cdkn1a in the sbds null background result in reversed neutropenia?

Response: We have created a zebrafish p21 KO zebrafish, and we are studying its phenotype. Preliminary data showed that it does not reverse the neutropenia or survival in the sbds KO background, (manuscript in preparation).

Comment 27: Performing qRT-PCR for all the transgenics (sbds^{+/+}, sbds^{-/-}, TgR126T;sbds^{+/+}, TgR126T;sbds^{-/-} 1copy and 2 copy) at Day1, Day2, Day5 and Day10 of development would be informative.

Response: Primarily because of the lower fecundity of the females and the lower numbers of females, this experiment would require at least 3-6 more months. While of some interest, we feel that our work is compelling and novel without these data.

September 13, 2023

RE: Life Science Alliance Manuscript #LSA-2022-01856-TR

Dr. Usua Oyarbide
Cleveland Clinic
Cancer Biology
9500 Euclid Avenue
Cleveland, Ohio 44195

Dear Dr. Oyarbide,

Thank you for submitting your revised manuscript entitled "SBDS R126T rescues survival of sbds^{-/-} zebrafish in a dose-dependent manner independently of Tp53". We would be happy to publish your paper in Life Science Alliance pending final revisions necessary to meet our formatting guidelines.

- please address Reviewer 1's remaining points
- please upload your main manuscript text as an editable doc file
- please upload your main figures as single files (one file per figure)
- please add a Summary Blurb/Alternate Abstract to our system
- please add the Twitter handle of your host institute/organization as well as your own or/and one of the authors in our system
- please label "Summary" in the manuscript text as Abstract
- please add your main figure legends to the main manuscript text after the references section

Figure checks:

- please add scale bars to Figure 2C

A. FINAL FILES:

B. MANUSCRIPT ORGANIZATION AND FORMATTING:

Sincerely,

Reviewer #1 (Comments to the Authors (Required)):

The authors have revised their manuscript to address most of the comments of the reviewers. They now evaluate the role of Tp53 in the phenotype of sbds null zebrafish by using a transcriptionally deficient Tp53 allele, tp53M214K. This loss-of-function Tp53 allele did not rescue neutropenia or overall survival of the sbds-null zebrafish. The authors conclude that the amount of what they describe as a 'hypomorphic' SBDS protein is important for zebrafish development.

Unfortunately, for me the novelty of the study is significantly reduced given previously published work in zebrafish (Provost et al Development (2012) 139 (17): 3232-3241. <https://doi.org/10.1242/dev.077107>) and mouse (Tourlakis et al. <https://doi.org/10.1371/journal.pgen.1005288>). The work by Provost et al should be cited.

The authors have still not addressed whether expression of the wild type human SBDS protein can rescue the sbds null zebrafish mutants. Without this essential control, it is not possible to claim that the SBDS-R126T mutant allele is hypomorphic in this context.

Typos.

The text is still replete with typos/errors/inconsistencies that obscure meaning. Please check the text and figure legends carefully. Some but not all the examples are given below.

a. Figure 1 legend: Please check this sentence and revise accordingly.

(E) Immunoblot showing protein levels in fish with 1 and 2 copies of the transgene comparing to wildtype sbds. of the ubiquitin promoter, cmlc2 green heart marker to aid screening of transgenesis.

b. Figure 2 legend

human SBDSR126T-please keep nomenclature consistent in text. Still written in multiple different formats as 'transgene Tg(SBDSR126T)' or 'Tg(hSBDSR126T)' or 'SBDSR126T' in italics or 'SBDSR126T' or SBDSR 126 (Figure 4).

c. Figure 6 legend: Crosses between sbds+/-, sometimes in text genotypes are italicized, sometimes not. No consistency. 'Crosses between sbds+/- (what?) with two copies with (please change to 'of') the transgene

d. Figure 2 legend: What is WKM, AUFS?

e. Eif6, EIF6 etc

Reviewer #2 (Comments to the Authors (Required)):

I have now gone through the author's response and the revised manuscript and I feel that the authors have sufficiently addressed my concerns.

Reviewer #1 (Comments to the Authors (Required)):

The authors have revised their manuscript to address most of the comments of the reviewers. They now evaluate the role of Tp53 in the phenotype of sbds null zebrafish by using a transcriptionally deficient Tp53 allele, tp53M214K. This loss-of-function Tp53 allele did not rescue neutropenia or overall survival of the sbds-null zebrafish. The authors conclude that the amount of what they describe as a 'hypomorphic' SBDS protein is important for zebrafish development.

Unfortunately, for me the novelty of the study is significantly reduced given previously published work in zebrafish (Provost et al Development (2012) 139 (17): 3232-3241. <https://doi.org/10.1242/dev.077107>) and mouse (Tourlakis et al. <https://doi.org/10.1371/journal.pgen.1005288>). The work by Provost et al should be cited.

Response: We have added the reference of Provost et al. in the discussion lines 266-288. We respectfully believe that our work is novel in multiple, significant ways: 1) we have used CRISPR instead of morpholinos so that our zebrafish model is a knockout and not a knockdown; 2) we have developed an organismal model that combines the loss of SBDS with the loss of function in TP53; and 3) our manuscript challenges the dogma that TP53 is solely responsible for the pathogenesis of Shwachman-Diamond syndrome (and may stimulate a re-examination of TP53's role in other marrow failure syndromes).

The authors have still not addressed whether expression of the wild type human SBDS protein can rescue the sbds null zebrafish mutants. Without this essential control, it is not possible to claim that the SBDS-R126T mutant allele is hypomorphic in this context.

Response: We chose the SBDS^{R126T} mutation because it is a disease-associated allele. It has been previously demonstrated that is a functional hypomorph by Alan Warren's and Johanna Rommen's groups (Finch et al 2011; Ball et al 2009; Tourlakis et al 2015). Following the suggestions of reviewer 1, we have changed the text in the manuscript including only SBDS^{R126T} and deleted the word hypomorphic. In our SBDS^{R126T} zebrafish model we have demonstrated that presence of the SBDS^{R126T} transgene causes an accumulation of Eif6 and upregulation of *cdkn1a*, key target of Tp53. Moreover, presence of the transgene did not rescue neutropenia at 10 dpf in the *sbds* KO background. We agree that it might be interesting to create

a transgenic line expressing the human wildtype SBDS, but our priority is to identify the Tp53-independent mechanism for the syndrome.

Typos.

The text is still replete with typos/errors/inconsistencies that obscure meaning. Please check the text and figure legends carefully. Some but not all the examples are given below.

Response: Thank you, we have corrected the typos.

a. Figure 1 legend: Please check this sentence and revise accordingly. (E) Immunoblot showing protein levels in fish with 1 and 2 copies of the transgene comparing to wildtype sbds. of the ubi-ubiquitin promoter, cmlc2 green heart marker to aid screening of transgenesis. We checked and changed this sentence.

b. Figure 2 legend human SBDSR126T-please keep nomenclature consistent in text. Still written in multiple different formats as 'transgene Tg(SBDSR126T)' or 'Tg(hSBDSR126T)' or 'SBDSR126T' in italics or 'SBDSR126T' or SBDSR 126 (Figure 4). We changed everything to SBDS^{R126T}.

c. Figure 6 legend: Crosses between sbds+/-, sometimes in text genotypes are italicized, sometimes not. No consistency. 'Crosses between sbds+/- (what?) with two copies with (please change to 'of') the transgene. We corrected this sentence to: Crosses between *sbds*^{+/-} zebrafish with two copies of the transgene

d. Figure 2 legend: What is WKM, AUFS? We added whole kidney marrow and absorbance units full scale to the legend.

e. Eif6, EIF6 etc. We have reviewed and changed them.

September 27, 2023

RE: Life Science Alliance Manuscript #LSA-2022-01856-TRR

Dr. Usua Oyarbide
Cleveland Clinic
Cancer Biology
9500 Euclid Avenue
Cleveland, Ohio 44195

Dear Dr. Oyarbide,

Thank you for submitting your Research Article entitled "SBDS R126T rescues survival of sbds^{-/-} zebrafish in a dose-dependent manner independently of Tp53". It is a pleasure to let you know that your manuscript is now accepted for publication in Life Science Alliance. Congratulations on this interesting work.

DISTRIBUTION OF MATERIALS:

Again, congratulations on a very nice paper. I hope you found the review process to be constructive and are pleased with how the manuscript was handled editorially. We look forward to future exciting submissions from your lab.

Sincerely,
